# Activation of LXR Receptors and Inhibition of TRAP1 Causes Synthetic Lethality in Solid Tumors

**DOI:** 10.3390/cancers11060788

**Published:** 2019-06-07

**Authors:** Trang Thi Thu Nguyen, Chiaki Tsuge Ishida, Enyuan Shang, Chang Shu, Elena Bianchetti, Georg Karpel-Massler, Markus D. Siegelin

**Affiliations:** 1Department of Pathology & Cell Biology, Columbia University Medical Center, New York, NY 10032, USA; tn2387@cumc.columbia.edu (T.T.T.N.); chiaki.tsuge@gmail.com (C.T.I.); cs485@cumc.columbia.edu (C.S.); eb2985@cumc.columbia.edu (E.B.); 2Department of Biological Sciences, Bronx Community College, City University of New York, Bronx, NY 10453, USA; es347@cumc.columbia.edu; 3Department of Neurosurgery, Ulm University Medical Center, 89081 Ulm, Germany; georg.karpel@gmail.com

**Keywords:** Gamitrinib (GTPP), Cholesterol, Sterol regulatory element-binding protein 2 (SREBP2)

## Abstract

Cholesterol is a pivotal factor for cancer cells to entertain their relentless growth. In this case, we provide a novel strategy to inhibit tumor growth by simultaneous activation of liver-X-receptors and interference with Tumor Necrosis Factor Receptor-associated Protein 1 (TRAP1). Informed by a transcriptomic and subsequent gene set enrichment analysis, we demonstrate that inhibition of TRAP1 results in suppression of the cholesterol synthesis pathway in stem-like and established glioblastoma (GBM) cells by destabilizing the transcription factor SREBP2. Notably, TRAP1 inhibition induced cell death, which was rescued by cholesterol and mevalonate. Activation of liver X receptor (LXR) by a clinically validated LXR agonist, LXR623, along with the TRAP1 inhibitor, gamitrinib (GTPP), results in synergistic reduction of tumor growth and cell death induction in a broad range of solid tumors, which is rescued by exogenous cholesterol. The LXR agonist and TRAP1 inhibitor mediated cell death is regulated at the level of Bcl-2 family proteins with an elevation of pro-apoptotic Noxa. Silencing of Noxa and its effector BAK attenuates cell death mediated by the combination treatment of LXR agonists and TRAP1 inhibition. Combined inhibition of TRAP1 and LXR agonists elicits a synergistic activation of the integrated stress response with an increase in activating transcription factor 4 (ATF4) driven by protein kinase RNA-like endoplasmic reticulum kinase (PERK). Silencing of ATF4 attenuates the increase of Noxa by using the combination treatment. Lastly, we demonstrate in patient-derived xenografts that the combination treatment of LXR623 and gamitrinib reduces tumor growth more potent than each compound. Taken together, these results suggest that TRAP1 inhibition and simultaneous activation of LXR might be a potent novel treatment strategy for solid malignancies.

## 1. Introduction

Targeting tumor metabolism for therapy has become “en vogue” over the last decade since tumor cells display a reprogrammed metabolism as compared to non-neoplastic cells [1,2]. This is exemplified by many alterations, including glycolysis, the citric acid cycle (TCA-cycle), fatty acid, phospholipid, and cholesterol/mevalonate synthesis [3,4,5]. These deviations set the stage for novel therapeutic strategies. Liver-x-receptor (LXR) are regulators of cholesterol levels in cells and, when activated, they lead to an up-regulation of a battery of different transporters and proteins, such as ABCA1, that elicit a reduction in cholesterol within the cells [6,7]. This acute efflux of cholesterol is compensated by enhanced cholesterol synthesis. LXR agonists, e.g., LXR623 and GW3965, have reached clinical testing and have gained traction in preclinical cancer research. In model systems of melanoma and glioblastoma, activation of LXR has resulted in extension of survival in several preclinical model systems [8].

The protein expression of the mitochondrial Hsp90 homologue, Tumor Necrosis Factor Receptor-associated Protein 1 (TRAP1), is increased in cancer cells [9,10]. It drives resistance toward programmed cell death [11] and is a regulator of glycolysis and oxidative phosphorylation in tumor cells. TRAP1 regulates oxidative phosphorylation at the level of complex II and interference with TRAP1 results in suppression of the oxygen consumption rate [12,13]. Targeting TRAP1 can be accomplished through several means. Initially, a cell-penetrating peptide was shown to inhibit TRAP1 and, later on, a class of molecules was developed, called gamitrinib (GTPP) [9]. Gamitrinib has been proven to be effective singly or in combination therapies, involving, but not limited to, PI3K inhibitors, TRAIL, and BH3-mimetics [14,15,16]. 

In this scenario, we provide evidence that targeting of TRAP1 by siRNA or gamitrinib inhibits the transcripts and protein levels of enzymes related to mevalonate and cholesterol synthesis by disrupting the stability of Sterol regulatory element-binding protein 2 (SREBP2) and that exogenous cholesterol and mevalonate can rescue from gamitrinib mediated cell death. In turn, we hypothesized that further reduction of cholesterol levels by activation of the liver-x-receptor should elicit a synthetic lethal interaction. We show that this strategy results in significant induction of apoptosis mediated by pro-apoptotic Noxa [17,18]. Mechanistically, we find that the combination treatment drives ER-stress with activation of activating transcription factor 4 (ATF4) in a synergistic manner. In turn, ATF4 drives up-regulation of Noxa, which facilitates intrinsic apoptosis. Importantly, our findings are validated in patient-derived xenograft models, which demonstrates the strongest anti-tumor efficacy in the combination treatment.

## 2. Results

### 2.1. TRAP1 Regulates Mevalonate and Cholesterol Biosynthesis Through Modulation of SREBP2

First, we assessed the expression of TRAP1 in the cell culture systems used in this study (Appendix A). In this vein, following interrogation of public databases, we found that TRAP1 levels are significantly increased in glioblastoma tissue vs. normal brain tissue and in colonic adenocarcinoma vs. normal colonic tissue, respectively (Appendix A). A transcriptome with subsequent gene set enrichment analysis demonstrated that stem-like glioblastoma (GBM) cells, NCH644, that were treated with a low concentration of TRAP1—inhibitor gamitrinib—displayed down regulated gene sets related to cholesterol and mevalonate biosynthesis (Figure 1a,b,d). We validated these genes by standard real-time PCR analysis, which showed suppression of transcripts of enzymes related to cholesterol biosynthesis (3-Hydroxy-3-Methylglutaryl-CoA Synthase 1(HMGCS1), 3-hydroxy-3-methyl-glutaryl-coenzyme A reductase (HMGCR), Mevalonate kinases (MVK), Phosphomevalonate Kinase (PMVK), Mevalonate Diphosphate Decarboxylase (MVD), 7-Dehydrocholesterol reductase (DHCR7)) (Figure 1c and Appendix A). Given that cholesterol biosynthesis is controlled by the master-regulator transcription factor, SREBP2, we hypothesized that both pharmacological and genetic TRAP1 inhibition might affect the levels of SREBP2. To test this hypothesis, we used both stem-like (NCH644) and established (U87) GBM cells, that were treated with gamitrinib in a time course experiment. Capillary electrophoresis demonstrated that as early as 8 hours, gamitrinib suppressed the levels of the full length (inactive) as well as the cleaved form of SREBP2 (Figure 1e,f). This effect became more prominent after 12 hours and 24 hours. Similar effects were seen in T98G and LN229 cells (Appendix A). To assess the impact on downstream targets of SREBP2 upon gamitrinib treatment, we determined the protein levels of the enzyme HMGCR, which catalyzes the committing step of cholesterol/mevalonate synthesis. In alignment with the reduction of the cleaved form of SREBP2, a suppression of HMGCR protein was detected (Figure 1e). To confirm that inhibition of TRAP1 is responsible for gamitrinib mediated suppression of SREBP2, we utilized two siRNAs targeting TRAP1 (Figure 1g and Appendix A). Akin to gamitrinib, silencing of TRAP1 results in down-regulation of both the full length and the cleaved form of SREBP2 as well as the downstream effector HMGCR (Figure 1g and Appendix A). Furthermore, we validated that SREBP2 suppression is sufficient to reduce cellular viability of U87 GBM cells by silencing SREBP2 (Appendix A).

Since the TRAP1 protein represents a chaperone protein, we hypothesized that SREBP2 is likely stabilized by TRAP1 and, once TRAP1 levels and/or activity decline, SREBP2 is subjected to degradation by the proteasome. Therefore, U87 GBM cells were treated with gamitrinib in the presence or absence of MG132. We found that the proteasome inhibitor, MG132, rescued gamitrinib mediated suppression of SREBP2 (Figure 1h). Given the involvement of enhanced proteasomal degradation upon gamitrinib treatment, we assessed the stability of SREBP2 protein (full length and cleaved form) in the presence or absence of TRAP1 inhibition and found that gamitrinib decreased the stability of SREBP2 (Figure 1i). To demonstrate that TRAP1 interacts with SREBP2, we immunoprecipitated SREBP2 and analyzed the expression levels of TRAP1 protein. We found that TRAP1 co-precipitated with SREBP2, which suggests that the two proteins interact. In contrast, a control immunoprecipitation with IgG did not show the presence of TRAP1, which supports the specificity of our findings (Figure 1j). To confirm that gamitrinib leads to lower levels of cholesterol levels, we determined total cholesterol levels in glioblastoma cells. As anticipated, gamitrinb lowered the total levels of cholesterol (Figure 1k).

Next, we assessed the impact of cholesterol on gamitrinib mediated reduction in viability and cell death induction in the context of rescue experiments. To this purpose, LN229 and U87 were treated with increasing concentrations of gamitrinib in the presence or absence of cholesterol. We found that LDL-cholesterol protected from gamitrinib mediated viability reduction and cell death induction (Figure 2a,b,d,e). Similarly, mevalonate rescued from gamitrinib mediated a reduction in cellular viability (Figure 2c). In contrast, as anticipated, sodium acetate did not reverse the cytotoxic effects of gamitrinib (Appendix A).

### 2.2. Activation of Liver-X-Receptors Coupled with Inhibition of Mitochondrial Matrix Chaperones Elicit Synergistic Induction of Apoptosis

Our findings provide a tumor-specific approach to suppress cholesterol biosynthesis by targeting TRAP1, which is up-regulated in tumor cells as compared to normal cells. However, since cholesterol might be taken up from the interstitium as part of a compensatory mechanism, we hypothesized that the effects of TRAP1 inhibition on tumor cell death might be further enhanced in the presence of compounds that facilitate cholesterol efflux. To this purpose, we combined gamitrinib with the clinical validated LXR agonist, LXR623, and assessed the cellular viability in several model systems of glioblastoma, including stem-like GBM and patient-derived xenograft cells. Equivocally, we found that the combination treatment of gamitrinib and LXR623 synergistically reduced tumor growth in all model systems tested, which includes stem-like GBM cells, but also other solid malignancies, such as melanoma (A375) and colonic carcinoma (HCT116) cells (Figure 2f–m and Appendix A). Neither single nor combination treatment of GTPP/LXR623 elicited a reduction of viability in astrocytes (Appendix A). 

To account for the cell death mechanisms by which gamitrinib and LXR623 elicit their biological effect, we analyzed the cells for features of apoptotic cell death. We found that gamitrinib and LXR623 synergistically enhance Annexin V staining in several model systems, preceded by loss of mitochondrial membrane potential and activation of the caspase cascade (Figure 3a–c,e,f and Appendix A). These findings suggest that the combination treatment acted by enhancing cell death/apoptosis. Human astrocytes did not respond to either single or combination treatment, which suggests a preferential tumor cell specific effect (Figure 3d and Appendix A). Next, we asked whether or not these cytotoxic effects may be rescued by cholesterol. As expected, cholesterol significantly attenuated the viability reduction and cell death induction by the combination treatment of gamitrinib and LXR623 (Figure 3g–j).

Given the role of apoptosis in gamitrinib and LXR623-mediated reduction in proliferation, we determined the protein expression levels of the intrinsic regulators of apoptosis. Our findings show that the combination treatment affected both anti-apoptotic and pro-apoptotic Bcl-2 family members. Particularly, we detected an increase in pro-apoptotic Noxa levels coupled with a modulation of Mcl-1 levels, which resulted in a change of the ratio of Noxa and Mcl-1. This favors a pro-apoptotic state (Figure 4a and Appendix A). Since Noxa binds to Mcl-1 and liberates BAK, we silenced the Noxa and BAK to assess their impact on cell death of the combination treatment (Figure 4c,d). As anticipated, suppression of Noxa and BAK attenuated apoptosis induction by the combination treatment (Figure 4b).

### 2.3. LXR Activation and Mitochondrial Matrix Chaperone Inhibition Induces Synergistic Endoplasmic Reticulum Stress with an Increase in ATF4

To better understand the upstream mechanisms by which the drug combination of gamitrinib and LXR623 work, we conducted an interaction analysis on our transcriptomic data. Our gene set enrichment analysis suggested that the combination treatment had higher enrichment for genes related to ER-stress than single drug treatments (Figure 5a,b). Consistently, our real-time PCR analysis of cells treated with the combination showed potent induction of ER-stress related mRNAs (Figure 5c). Next, we validated these findings on the protein level. As early as 7 hours into the process, we found an increase of ER-stress markers, GRP78, ATF4, and CHOP, which was most pronounced in the combination treatment (Figure 5d and Appendix A). The up-regulation of ATF4 was accompanied by an enhanced phosphorylation of elF2α, which is mediated by the PERK-kinase (Appendix A). Next, we tested whether or not exogenous cholesterol may rescue LXR623 and gamitrinib-mediated induction of the integrated stress response and found that cholesterol administration potently suppressed the up-regulation of ATF4 elicited by the combination treatment (Figure 5e).

Given the fact that ATF4 is capable of regulating the expression of Noxa, we determined whether or not the observed enhanced Noxa expression elicited by the combination treatment is related to ATF4 and its upstream regulator PERK. To this purpose, we silenced the expression of PERK by using two different oligonucleotides in LN229 cells (Figure 5f,g). Thereafter, cells were treated with gamitrinib, LXR623, and the combination of both. We found that ATF4 and CHOP were potently up-regulated by the combination treatment in the presence of non-targeting siRNA. However, PERK targeting siRNAs almost completely suppressed this increase, which suggests that the combination treatment acts, in part, through the PERK-elF2α-ATF4 pathway (Figure 5f,g). Lastly, we assessed the importance of ATF4 for the up-regulation of Noxa upon treatment with gamitrinib and LXR623. We found that silencing of ATF4 attenuates gamitrinib and LXR623 mediated up-regulation of both CHOP and Noxa, which suggests that ATF4 is pivotal in the increase of Noxa elicited by the combination treatment (Figure 5h). These findings suggest that the combination treatment works in part by up-regulation of ATF4.

### 2.4. The Combination Treatment of Gamitrinib and LXR623 Elicits Anti-Cancer Activity in Conventional and Patient-Derived Xenograft Model Systems

The translation from an in vitro system into an in vivo setting is a major challenge in cancer research. To ensure that our combination treatment is relevant and effective in vivo, we used the current state of the art model system, which is patient-derived xenografts (PDX). The GBM12 PDX was used and, after injection and establishment of tumors, animals were randomly assigned to four treatment groups (A: Vehicle, B: gamitrinib (GTPP), C: LXR623, and D: gamitrinib and LXR623). While gamitrinib displayed little effects on tumor growth, we detected a growth reduction by LXR623. However, the combination treatment of gamitrinib and LXR623 was most potent to reduce tumor growth (Figure 6a–c). Histopathological evaluation of the individual groups demonstrated that the combination treatment induced broad areas of necrosis, whereas smaller areas of necrosis was noted in vehicle or single treatment (Figure 6g). Less mitotic figures were appreciated in the combination treatment (Figure 6j). These observations are in keeping with the effects on tumor growth. We noted the strongest increase in TUNEL—staining in the combination treatment (Figure 6h,k). In contrast, the proliferation rate was most potently suppressed in the combination treatment as assessed by Ki67-staining (Figure 6i,l). Given the potency of this treatment, we evaluated toxicity by organ histology. While the combination treatment had drastic effects on tumor growth with massive necrosis, organs such as brain, liver, kidney, lung, heart, intestine, pancreas, and spleen were not affected by the combination treatment and no significant weight losses were noted during the treatments. This suggests a favorable toxicity profile (Appendix A).

We extended our in vivo findings to another model, which involves glioblastoma stem-like cells, NCH644. Four treatment groups were assigned after the establishment of tumors. While we noted some growth suppression elicited by gamitrinib, LXR623 alone demonstrated little potency. Notably, the combination treatment suppressed tumor growth significantly when compared to vehicle or single treatments, which is in keeping with results obtained in the PDX model (Figure 6d–f).

While we focused mostly on glioblastoma, it was tempting to determine as to whether or not synergism of gamitrinib and LXR623 would also be true in other tumor in vivo systems as well. To this purpose, we utilized a colon cancer xenograft model, which involves the well-known HCT116 cells. Akin to the other model systems, treatment groups were assigned and drugs were administered accordingly. While the single treatment did not result in substantial growth reduction, the combination treatment reduced tumor size as compared to vehicle and single treatments (Appendix A).

Collectively, our results suggest that activation of LXR along with TRAP1 inhibition might be a novel strategy to combat recalcitrant malignancies.

## 3. Discussion

Targeting metabolic pathways for cancer therapy has become an attractive pursuit in the last decade [19,20]. While it is undoubtedly true that genetic alterations are drivers of carcinogenesis, molecular alterations impact tumor cell metabolism and, thereby, create targetable vulnerabilities. Cholesterol has been shown to be pivotal for cancer growth since it participates in many fundamental processes to maintain viability. To entertain high growth rates, constant synthesis, and/or uptake of this molecule has to be ensured. Cholesterol synthesis may be blocked by classical inhibitors, such as HMG-CoA-reductase inhibitors [21] that are standard of care for hypercholesterinemia and/or coronary artery disease. However, these compounds are not tumor-specific blockers of cholesterol synthesis and they bear serious side effects, involving muscular dystrophy of the lower extremities that results in severe cases in the inability to exercise or, in some egregious cases, normal ambulating. Therefore, it would be desirable to identify an approach that would permit suppression of cholesterol synthesis preferentially in tumor cells and not normal cells. In this study, we have identified a novel strategy to inhibit cholesterol/mevalonate pathways by interference with TRAP1. TRAP1 is significantly up-regulated in cancer cells and, therefore, targeting this molecule provides a “therapeutic window” in the treatment of malignant tumors. Our findings in this scenario demonstrate that TRAP1 regulates cholesterol synthesis, which we have demonstrated both genetically as well as pharmacologically. This involves the compound gamitrinib, which is soon to be launched for clinical testing. A significant feature about this finding is that it enables a “tumor specific” approach to interfere with cholesterol synthesis since, as mentioned, the target TRAP1 is increased in tumor cells. The observation that TRAP1 is implicated in the regulation of tumor cell cholesterol synthesis is novel and in support of the notion that TRAP1 in certain contexts (as in the current one) acts as an oncogene. It is also important to note that cholesterol was able to rescue from gamitrinib-mediated cell death, whereas acetate had no effect. This observation supports the importance of cholesterol in TRAP1 inhibition-mediated cell death. Other groups have highlighted the importance of the mevalonate synthesis pathway in glioblastoma cells and suggested that brain tumor initiating cells [22] activate this pathway, in part, through the transcription factor, c-Myc [23]. Elegant chromatin immunoprecipitation studies confirm binding of c-Myc to the promoter regions of several genes, involved in both cholesterol and mevalonate synthesis, including PMVK and HMGCR. Consequently, silencing of c-Myc suppressed the expression of these transcripts and suggests that c-Myc is likely be a regulator of mevalonate/cholesterol synthesis in brain tumor-initiating cells. Our study is distinct from this work by the described mechanisms regarding how cholesterol synthesis is maintained in tumor cells. It is acknowledged that these signaling pathways might interact and enhance each other since both TRAP1 and c-Myc are simultaneously up-regulated in brain tumor-initiating cells [24].

Mechanistically, TRAP1 promotes high levels of SREBP2 [25,26] by inhibiting its proteasomal degradation. SREBP2 is a master-regulator transcription factor of cholesterol synthesis that controls the expression levels of cholesterol synthesis enzymes [27]. Prior work by others suggest that SREBP2 is central to maintain survival of glioblastoma cells, especially when it is simultaneously inhibited along with SREBP1 [28]. Dual inhibition of SREBP1 and SREBP2 resulted in enhanced ER-stress, which was rescued by BSA-oleate. Similarly, in our study, cholesterol was able to rescue the cytotoxic effects of both gamitrinib and the combination treatment, which supports the notion that cholesterol depletion is potentially implicated in the cell death mechanism of these reagents and/or pathways. Additional work is necessary when the effects of TRAP1 are only limited on SREBP2. For instance, it is conceivable that TRAP1 might also regulate the levels of SREBP1, which regulates fatty acid synthesis. Moreover, TRAP1 may interact with other master regulator transcription factors that globally regulate tumor cell metabolism, e.g., c-Myc. Akin to cholesterol metabolism, fatty acid synthesis is a cornerstone for tumor growth. Deprivation of fatty acid synthesis will dampen tumor viability [29]. Recently, a drug compound, called ND-646, was shown to dramatically suppress tumor growth by inhibiting the regulatory enzyme of fatty acid synthesis, acetyl-coa-carboxylase (ACC) [29]. 

Since single treatment fall often short of expectations, we hypothesized that the cytotoxic effects of gamitrinib may be further enhanced through the addition of LXR agonists. While such a strategy has not been suggested earlier, we found that this combination treatment was potent to elucidate synergistic growth reduction by activation of enhanced ER-stress with up-regulation of pro-apoptotic Noxa, which was mediated by the PERK-ATF4-eif2 axis. Consequently, silencing of Noxa attenuated the effects of the combination treatment. Our prior study suggested before that gamitrinib itself may modulate Noxa and its anti-apoptotic binding partner Mcl-1. However, the current combination treatment suggests that the increase in Noxa by gamitrinib is significantly further enhanced in the presence of LXR agonists. It is also important to note that the upstream mechanism (ATF4) are significantly less activated by gamitrinib/LXR623 in the presence of cholesterol, which highlights the importance of cholesterol in the cell death mechanism. It also appears likely that the induction of Noxa occurs independent of TP53 since many of the tested cell lines contain mutated TP53, such as LN229. This is relevant since a significant proportion of gliomas harbor mutation in TP53. 

Given these promising in vitro findings, we tested the combination treatment in conventional and PDX-model systems and found that the combination treatment is superior to single treatments without induction of detectable histological organ toxicity, which reinforces this strategy for a potential application to patients. To date, these findings are unique since the combination treatment of gamitrinib and LXR agonists has never been tested in any animal model before. Future studies are warranted to assess the role of cholesterol in vivo in the context of single or combination drug treatments. It would be anticipated that a low cholesterol diet may enhance the efficacy of gamitrinib, whereas a high-cholesterol diet may attenuate its effects.

## 4. Material and Methods

### 4.1. Reagents and Antibodies

Gamitrinib (GTPP) was kindly provided by Dr. Dario Altieri (Wistar Institute, Philadelphia, PA, USA). LXR623 were purchased from Selleckchem (Houston, TX, USA). Low Density Lipoprotein from Human Plasma (LDL) was purchased from Thermo Fisher (LDL-cholesterol). A 10 mM working solution in dimethylsulfoxide (DMSO) was prepared for all reagents prior to storage at –20 °C. Final concentrations of DMSO were below 0.1% (v/v). The following antibodies were used: Mcl-1 (1:500; CST: Cell Signaling Technology, Danvers, MA, USA), BAK (1:500; CST), Bcl-2 (1:500; CST), BIM (1:500; CST), Bcl-xL (1:500; CST), Noxa (1:500, clone 114C307; Calbiochem, Burlington, MA, USA), β-actin (1:000, clone AC15; Sigma Aldrich, St. Louis, MO, USA), ATF4 (1:500; CST), GRP78 (1:500, CST), PERK (1:500, clone D11A8; CST), CHOP (1:500, CST), elF2α (1:25; CST), p-elF2α(1:25; CST), SREBP2 (1:700, Novus Biologicals, Centennial, CO, USA), TRAP1 (1:1000, BD Bioscience, San Jose, CA, USA), HMGCR (1:25, Abcam, Cambridge, MA, USA), Vinculin (1:500; Abcam), PAPRP (1:500; CST), Caspase 9 (1:500; CST), and secondary HRP-linked antibodies were purchased from Santa Cruz Biotechnology Inc. (Dallas, TX, USA).

### 4.2. Cell Cultures and Transfection

The LN229, U87, T98G, HCT116, WC62, GBM12, and GBM22 cells were cultured in DMEM containing 10% FBS (Gemcell, New York, NY, USA), primocin (InvivoGen, San Diego, CA, USA), and in a humidified air (5% CO_2_) at 37 °C. For the experiments, cells were cultured in DMEM containing 1.5% FBS (non-lipoprotein deficient) and primocin (InvivoGen). In selected instances, lipoprotein deficient serum was employed. NCH644, NCH421k, and NCH690 glioma stem-like cells were cultured in MG-43 medium (CLS, Heidelberg, Germany) for both maintenance and experiments [30]. A375 cells were cultured in RPMI medium 1640 containing 10% FBS (Gemcell), primocin (InvivoGen). For the experiments, cells were cultured in RPMI medium 1640 containing 1.5% FBS (non-lipoprotein deficient) and primocin (InvivoGen). Cells were transfected using either Lipofectamine RNAiMAX or Lipofectamine 3000 (Invitrogen) and harvested after 72 h after transfection. siTRAP1-2, siSREBP2, siPMAIP1 (siNoxa), and siBAK were purchased from Ambion. Non-targeting siRNA-pool (siNT), siTRAP1 (pool), and siATF4 were purchased from Dharmacon (Lafayette, CO, USA). siPERK-1 and siPERK-2 were purchased from Cell Signaling.

### 4.3. Total Cholesterol Measurement

Cells were seeded in a six-well plate with 1 × 10^5^ cells/well and allowed to attach overnight. Cells were treated with reagents or corresponding solvents in the medium containing 1.5% lipoprotein deficient serum on the following day and incubated for 48 h. Cells were harvested and washed once in PBS. The cells were suspended in 100 µL of 1× Amplex Red reaction buffer (0.1 M Potassium phosphate, 0.05 M NaCl, 5 mM Cholic Acid, 0.1% Triton X-100, and pH 7.4) (Thermo Fisher, Waltham, MA, USA) and vortexed. To examine cholesterol levels, the Amplex Red Cholesterol Assay Kit was used in accordance with the manufacturer’s instructions (Thermo Fisher).

### 4.4. Cell Viability Assays

Cells were seeded in a 96-well flat bottomed plate and allowed to attach overnight. Cells were treated with reagents or corresponding solvents in the medium containing 1.5% FBS on the following day and incubated for 72 h. To examine cellular proliferation, CellTiter-Glo^®^ assays were performed, according to the manufacturer’s instructions (Promega, Madison, WI, USA). For cholesterol rescue experiments, cells were cultured in 1.5% lipoprotein deficient serum (Thermo Fisher).

### 4.5. Measurement of Apoptosis and Mitochondrial Membrane Potential

Cells were seeded in a 12-well plate and allowed to attach overnight. Cells were treated with reagents or corresponding solvents in the medium containing 1.5% FBS on the following day and incubated for the indicated time. To detect the apoptosis and necrosis, cells were labeled with FITC Annexin V/propidium iodide, according to the manufacturer’s instructions (BD Biosciences). To detect DNA fragmentation, PI staining was performed according to the manufacturer’s instructions (Cell Signaling Technology). To measure loss of mitochondrial membrane potential, TMRE (tetramethylrhodamine ethyl ester perchlorate) staining was performed according to the manufacturer’s instructions (Cell Signaling Technology). All samples were read on an LSRII flow cytometry (Becton–Dickinson, NJ, USA) and the data were analyzed with FlowJo software (version 8.7.1, Tree Star, Ashland, OR, USA).

### 4.6. Western Blot Analysis and Capillary Electrophoresis on Wes Instrument (ProteinSimple)

Cells were collected and lysed in cell extraction buffer (Invitrogen, Carlsbad, CA, USA) containing 1mM PMSF and 1× Protease and Phosphatase Inhibitor Cocktail (Thermo Fisher Scientific). Cell extracts were run on a 4–12% SDS PAGE gel (Invitrogen). The proteins were transferred to a PVDF membrane and the membrane was blocked with 5% milk in TBST (0.1% Tween 20) as well as probed with target antibodies. The blot was also probed with a β-actin or Vinculin as a loading control. The Western blots were acquired by using the Azure (C300) imaging system (Azure Biosystems, Dublin, CA, USA). Capillary electrophoresis was performed on the Wes instrument, according to the manufacturer’s instructions (ProteinSimple, CA, USA). The whole blot information can be accessed in Appendix A.

### 4.7. Real-Time PCR Analysis

To analyze the expression of target mRNAs, total RNA was isolated using the miRNAeasy Mini Kit (QIAGEN, Germantown, MD, USA) and reverse-transcribed to cDNAs using cDNA synthesis kit (OriGene, Rockville, MD, USA). The generated cDNA was amplified using power SYBR green RT-PCR reagents kit (Applied Biosystems, Foster City, CA, USA). The reaction was run at 95 °C for 10 min, which was followed by 40 cycles of 95 °C for 15 s, 60 °C for 30 s, and 72 °C for 30 s, on a qPCR Instrument (Quantabio, Beverly, MA, USA). The relative transcript expression levels were measured by quantitative real-time PCR using the SYBR Green-based method. The average fold changes were calculated based on 18S in the threshold cycle (Cq). All the primer sequences are in Table 1. 

### 4.8. Transcriptome Analysis

Microarray and subsequent gene set enrichment analysis were performed as described earlier [31]. The experiment used in this study were deposited at GEO: GSE104272.

### 4.9. Subcutaneous Xenograft Models

1 × 10^6^ GBM12 cells or NCH644 cells suspended 1:1 in Matrigel^®^ Matrix (Corning, NY, USA) were implanted subcutaneously into the flanks of 6-week-old to 8-week-old Nu/Nu mice. Tumors were measured with a caliper and sizes were calculated according to the standard formula: (length × width^2^) × 0.5. Treatment was performed intraperitoneally 3 times a week for 3 weeks. For intraperitoneal application, GTPP and LXR623 were dissolved in 10% DMSO, 32% Cremophor EL (SIGMA, St. Louis, MO, USA), 8% Ethanol (Pharmco-Aaper, Brookfield, CT, USA), and 50% PBS. Representative tumors were harvested, fixed in formalin, and, thereafter, embedded in paraffin. Afterward, sections were cut and were stained for hematoxylin and eosin (H.E.-stain) by standard laboratory procedures. In addition, immunohistochemistry stains for Ki67 (Agilent, GA62661-2) and TUNEL (terminal deoxynucleotidyl transferase dUTP nick end labeling) were performed.

### 4.10. Statistical Analysis

Statistical significance was assessed by two-tailed Student’s *t*-test using Prism version 5.04 (GraphPad, La Jolla, CA, USA). A *p* ≤ 0.05 was considered statistically significant. For multiple comparisons, we utilized ANOVA. The CompuSyn software (Version 3.0.1, ComboSyn, Inc., Paramus, NJ, USA, www.combosyn.com) was used for the drug combination analysis including the calculation of the combination index (CI). A CI < 1 was considered as synergistic, a CI = 1 as additive, and a CI > 1 as antagonistic.

### 4.11. Study Approval

All procedures were in accordance with Animal Welfare Regulations and approved by the Institutional Animal Care and Use Committee at the Columbia University Medical Center (AC-AAAQ4443, 7 June 2016).

## 5. Conclusions

Inhibition of TRAP1 along with activation of LXR results in synergistic growth reduction of tumor cells in vitro and in vivo. Thus, such a treatment strategy might be worthwhile for clinical testing in the future.

## Figures and Tables

**Figure 1 cancers-11-00788-f001:**
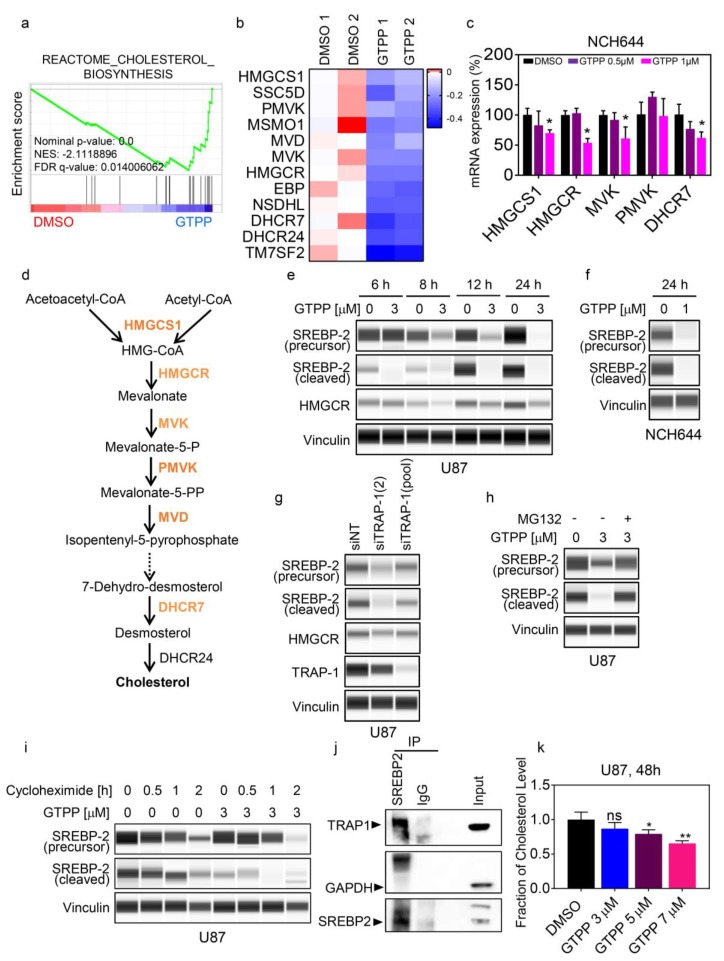
Tumor Necrosis Factor Receptor-associated Protein 1 (TRAP1) regulates cholesterol biosynthesis. (**a**) NCH644 cells were treated with 1 µM gamitrinib (GTPP) for 24 hours. Transcriptome and gene set enrichment analysis was performed. Shown is an enrichment plot. NES: normalized enrichment score. (**b**) Shown is a heat map of transcriptome and gene set enrichment analysis from NCH644 cells treated with 1 µM GTPP and control cells. (**c**) Real time PCR analysis of NCH644 cells were treated with 1 µM GTPP for 24 hours. Shown are means and SD (*n* ≥ 4). (**d**) Scheme of cholesterol biosynthesis. (**e**) U87 cells were treated with 3 µM GTPP for the indicated times. Thereafter, lysates were collected and analyzed for the expression of SREBP2 (precursor), SREBP2 (cleaved), and HMGCR. Vinculin is used as loading control. (**f**) NCH644 cells were treated with 1 µM GTPP for 24 hours. Thereafter, lysates were collected and analyzed for the expression of Sterol regulatory element-binding protein 2 (SREBP2) (precursor) and SREBP2 (cleaved). (**g**) U87 cells were transfected with siNT, siTRAP1-2, or siTRAP1 (pool) for 72 h. Lysates were collected and analyzed for the expression of SREBP2 (precursor), SREBP2 (cleaved), TRAP1, and HMGCR. (**h**) U87 cells were treated with 3 µM GTPP in the absence or presence of MG132. Thereafter, lysates were collected and analyzed for the expression of SREBP2 (precursor) and SREBP2 (cleaved). (**i**) U87 cells were treated with Cycloheximide in the absence or presence of 3 µM GTPP. Lysates were collected and analyzed for the expression of SREBP2 (precursor) and SREBP2 (cleaved). (**j**) U87 cells were collected and protein lysates were prepared and immunoprecipitated with an antibody against SREBP2 or IgG control. Standard Western blotting was performed (immunoprecipitation and the corresponding inputs) with antibody against TRAP1, SREBP2, and GAPDH. The arrows highlight the specific protein bands. (**k**) U87 cells were treated with increasing concentrations of GTPP for 48 hours. Thereafter, lysates were collected and analyzed for total cholesterol levels. * *p* < 0.05; ** *p* < 0.01; ***/**** *p* < 0.001; n.s: not significant.

**Figure 2 cancers-11-00788-f002:**
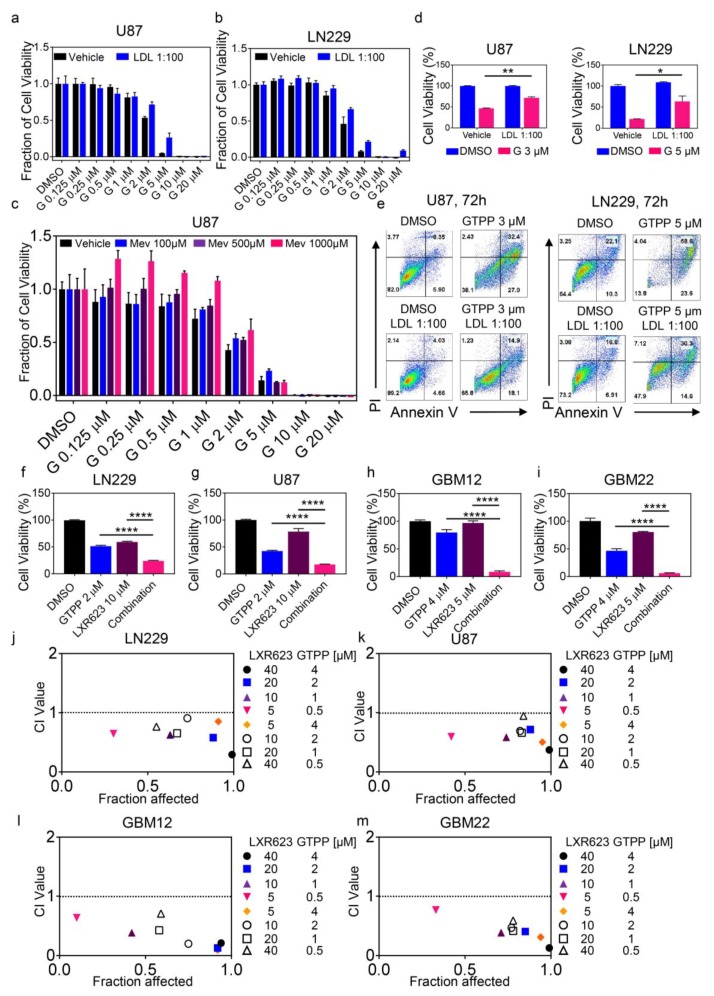
TRAP1 inhibition-mediated cell death is rescued by cholesterol or mevalonate and further enhanced by LXR agonists. (**a**,**b**) U87 and LN229 cells were treated with increasing concentration of GTPP in the presence or absence of cholesterol for 72 hours. Thereafter, cellular viability was analyzed and statistical analysis was performed. Shown are means and SD (*n* ≥ 4). G: GTPP and LDL: Low Density Lipoprotein. (**c**) U87 cells were treated with increasing concentration of GTPP in the presence or absence of mevalonate (Mev) for 72 hours. Thereafter, cellular viability was analyzed and statistical analysis was performed. Shown are means and SD (*n* ≥ 4). (**d**,**e**) U87 and LN229 cells were treated with indicated concentration of GTPP in the presence or absence of cholesterol for 72 hours. Thereafter, cells were stained with annexin V/propidium iodide and analyzed by multi-parametric flow cytometry. From this assay, cell viability was calculated. Shown are means and SD (*n* ≥ 3). (**f**–**i**) LN229, U87, GBM12, and GBM22 cells were treated with the indicated concentrations of GTPP, LXR623, or the combination of both for 72 hours. Thereafter, cellular viability was analyzed and statistical analysis was performed. Shown are means and SD (*n* ≥ 4). (**j**–**m**) CI (combination index) value indicates as to whether the drug combination is either additive (CI value = 1.0), synergistic (CI value < 1.0), or antagonistic (CI value > 1.0). The dotted line represents additivity with respect to the combination treatment. * *p* < 0.05; ** *p* < 0.01; ***/**** *p* < 0.001.

**Figure 3 cancers-11-00788-f003:**
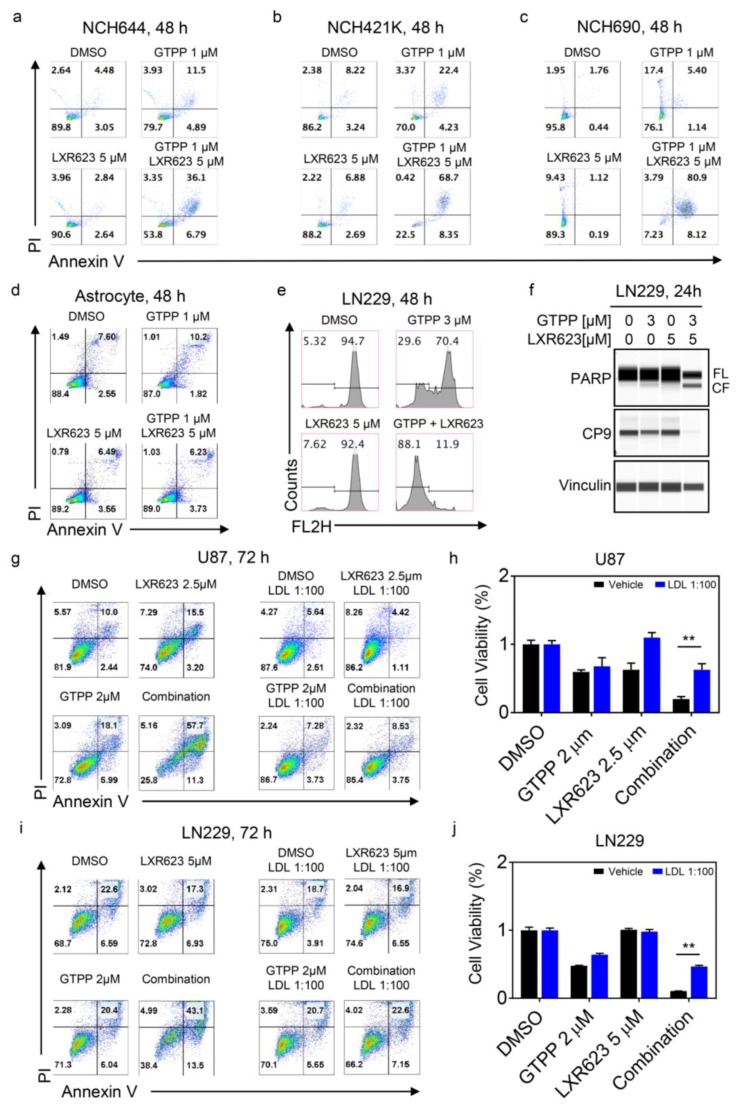
The combination treatment of LXR623 and gamitrinib elicits synergistic cell death and is rescued by cholesterol. (**a**–**c**) NCH644, NCH421K, and NCH690 stem-like GBM cells were treated with 1 µM GTPP, 5 µM LXR623, or the combination of both for 48 hours. Thereafter, cells were stained with annexin V/propidium iodide and analyzed by multi-parametric flow cytometry. (**d**) Human astrocytes were treated with 1 µM GTPP, 5 µM LXR623, or the combination of both for 48 hours, stained with annexin V/propidium iodide, and analyzed by multi-parametric flow cytometry. (**e**) LN229 (48 hours of treatment) were treated with 3 µM GTPP, 5 µM LXR623, or the combination of both, stained with TMRE and analyzed by flow cytometric analysis for dissipation of mitochondrial membrane potential. (**f**) LN229 cells were treated with 3 µM GTPP, 5 µM LXR623, or the combination of both. Whole protein lysates were collected and subjected for capillary electrophoresis (Wes, Proteinsimple, CA, USA) for the expression/cleavage of PARP, total caspase-9 (CP9), and Vinculin. (**g**) U87 cells were treated with 2 µM GTPP, 2.5 µM LXR623, or the combination of both in the presence or absence of cholesterol for 72 hours. Thereafter, cells were stained with annexin V/propidium iodide and analyzed by multi-parametric flow cytometry. (**h**) U87 cells were treated with 2 µM GTPP, 2.5 µM LXR623, or the combination of both in the presence or absence of cholesterol for 72 hours. Thereafter, cellular viability was analyzed and statistical analysis was performed. Shown are means and SD (*n* ≥ 3). (**i**) LN229 cells were treated with 2 µM GTPP, 5 µM LXR623, or the combination of both in the presence or absence of cholesterol for 72 hours. Thereafter, cells were stained with annexin V/propidium iodide and analyzed by multi-parametric flow cytometry. (**j**) LN229 cells were treated with 2 µM GTPP, 5 µM LXR623, or the combination of both in the presence or absence of cholesterol for 72 hours. Thereafter, cellular viability was analyzed and statistical analysis was performed. Shown are means and SD (*n* ≥ 3). ** *p* < 0.01.

**Figure 4 cancers-11-00788-f004:**
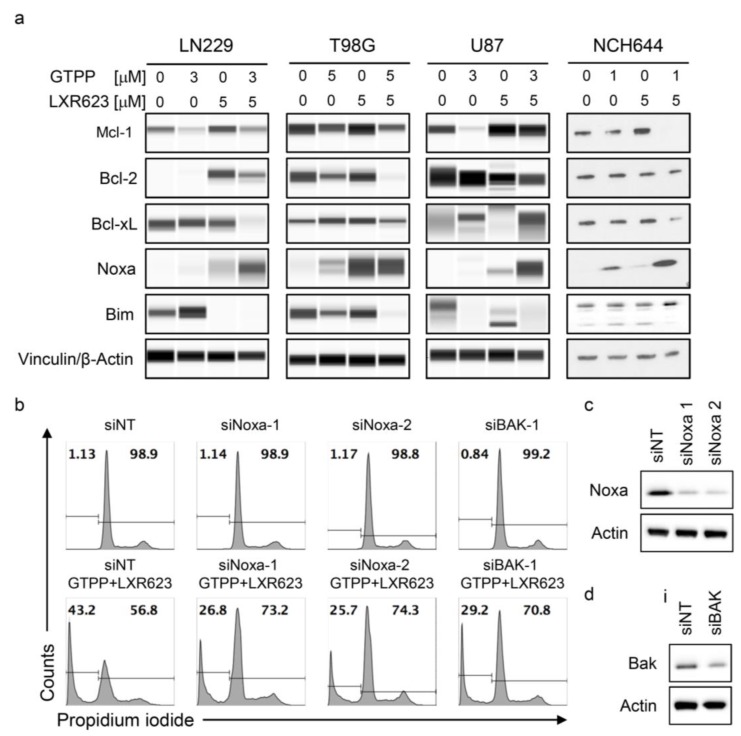
Gamitrinib and LXR623 regulate the expression of Bcl-2 family of proteins. (**a**) LN229, T98G, U87, and NCH644 cells were treated with the indicated concentrations of GTPP, LXR623, or combination of both for 72 hours. Thereafter, protein lysates were prepared and analyzed for the expression of Mcl-1, Bcl-2, Bcl-xL, Noxa, Bcl-2-like protein 11 (BIM), and Vinculin. (**b**) LN229 were transfected with siNT, siNoxa 1, siNoxa 2, or BAK-siRNA for 72 hours. Thereafter, cells were treated with the combination treatment of 3 µM GTPP and 20 µM LXR623 for another 24 hours. Thereafter, cells were harvested, fixed, stained with propidium iodide, and analyzed by flow cytometry for DNA fragmentation. (**c**,**d**) LN229 were transfected as described in B. Thereafter, whole cell protein lysates were collected and analyzed by capillary electrophoresis for the expression of Noxa, BAK, and Vinculin.

**Figure 5 cancers-11-00788-f005:**
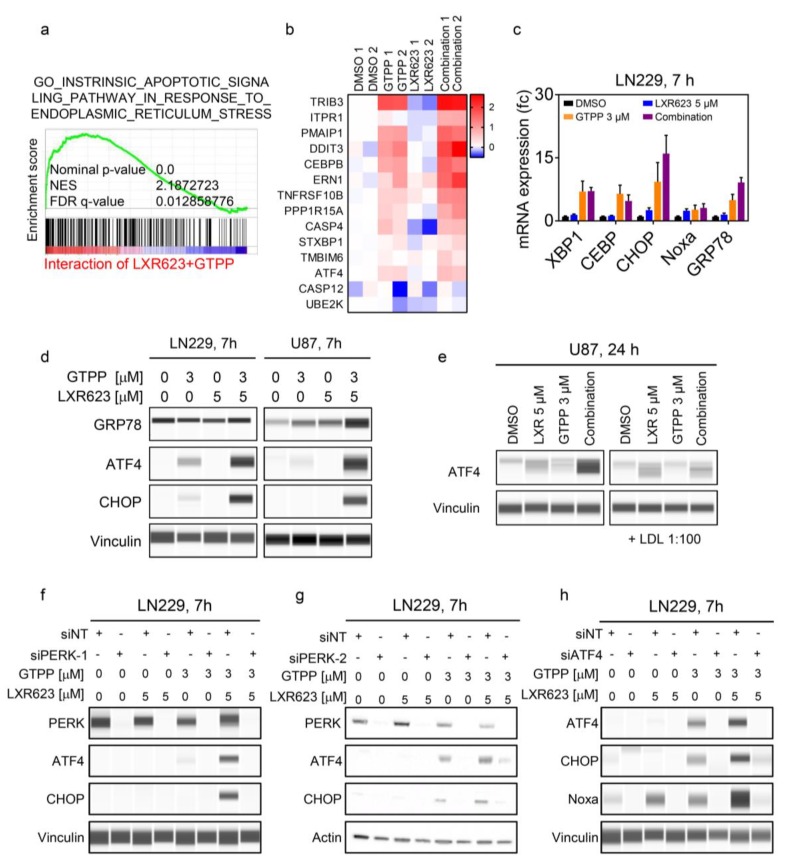
Gamitrinib and LXR623 elicit synergistic activation of the integrated stress response to drive the expression of pro-apoptotic Noxa. (**a**) NCH644 cells were treated with 1 µM GTPP and 5 µM LXR623 for 24 hours. Transcriptome and gene set enrichment analysis were performed with regards to the interaction of the combination treatment. Shown are enrichment plots. NES: normalized enrichment score. (**b**) Shown is a heat map of transcriptome and gene set enrichment analysis from NCH644 cells treated with vehicle, 1 µM GTPP, 5 µM LXR623, and the combination of both. (**c**) Real-time PCR analysis of LN229 cells were treated with 3 µM GTPP, 5 µM LXR623, or the combination of both for 24 hours. Shown are means and SD (*n* ≥ 4). (**d**) U87 and LN229 cells were treated with 3 µM GTPP, 5 µM LXR623, or the combination of both for 7 hours. Protein lysates were prepared and analyzed for the expression of GRP78, ATF4, CHOP, and Vinculin. (**e**) U87 cells were treated with 3 µM GTPP, 5 µM LXR623, or the combination of both in the presence or absence of cholesterol for 24 hours. Thereafter, protein lysates were prepared and analyzed for the expression of ATF4 and Vinculin. (**f**,**g**) LN229 were transfected with siNT, siPERK 1, or siPERK 2 for 72 hours. Thereafter, cells were treated with 3 µM GTPP, 5 µM LXR623, or the combination of both for another 24 hours. Protein lysates were prepared and analyzed for the expression of PERK, ATF4, CHOP, and Vinculin. (**h**) LN229 were transfected with siNT, siATF4 for 72 hours. Thereafter, cells were treated with 3 µM GTPP, 5 µM LXR623, or the combination of both for another 24 hours. Protein lysates were prepared and analyzed for the expression of ATF4, CHOP, Noxa, and Vinculin.

**Figure 6 cancers-11-00788-f006:**
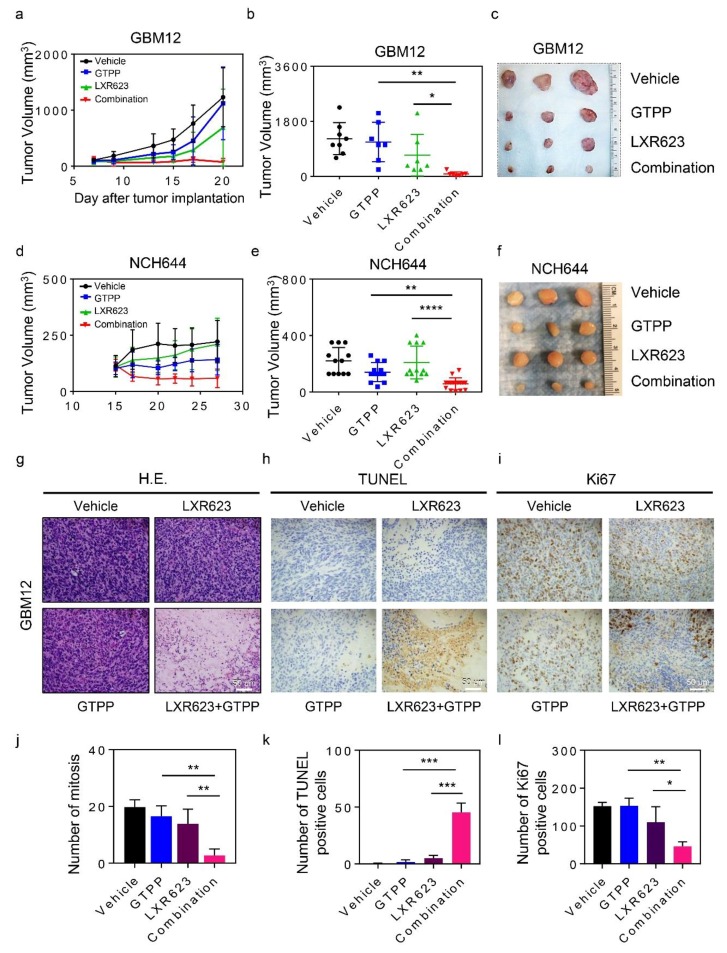
The combination treatment of gamitrinib and LXR623 elicits synergistic tumor growth in vivo. (**a**) GBM12 Patient-Derived Xenograft (PDX) tissue was implanted subcutaneously. After tumor formation, groups were formed. Animals were treated intraperitoneally with vehicle, LXR623 (200 mg/kg), GTPP (5 mg/kg), or both agents (six treatments provided during each size measurement). Tumor growth curves show the development of tumor size for each treatment group. Shown are means and SD (*n* ≥ 8). (**b**) Scatter plots display the quantitative representation of the tumor size among the different treatments toward the end of the experiment. Shown are means and SD (*n* ≥ 8). (**c**) Gross images of the in vivo xenograft experiments described in (**a**). (**d**) 1 × 10^6^ NCH644 cells were implanted subcutaneously. After tumor formation, groups were formed. Animals were treated intraperitoneally with vehicle, LXR623 (200 mg/kg), GTPP (5 mg/kg), or both agents (six treatments provided during each size measurement). Tumor growth curves show the development of tumor size for each treatment group. Shown are means and SD (*n* ≥ 12). (**e**) Scatter plots display the quantitative representation of the tumor size among the different treatments toward the end of the experiment. Shown are means and SD (*n* ≥ 12). (**f**) Gross images of the in vivo xenograft experiments described in (**a**). (**g**) Related to the GBM12 xenograft model, representative histopathological images (hematoxylin and eosin stain) are shown from each treatment group. (**h**,**i**) Sections from the same treatment groups as in G were stained with TUNEL or Ki67, respectively. Representative photographs were taken and are shown. Scale bar 50 µM. (**j**–**l**) Quantifications for mitosis (**j**), number of TUNEL (**k**), and Ki67 (**l**) positive cells are provided from the GBM12 tumor groups. Several high-power fields were counted (means and SD). * *p* < 0.05; ** *p* < 0.01; ***/**** *p* < 0.001.

**Table 1 cancers-11-00788-t001:** Primers for real-time PCR.

Gene	Forward Sequence	Reserve Sequence
HMGCS1	AAGTCACACAAGATGCTACACCG	TCAGCGAAGACATCTGGTGCCA
HMGCR	GACGTGAACCTATGCTGGTCAG	GGTATCTGTTTCAGCCACTAAGG
MVK	GGAAAGTGGACCTCAGCTTACC	GCTTCTCCACTTGCTCTGAGGT
PMVK	GCCTTTCGGAAGGACATGATCC	ACTCTCCGTGTGTCACTCACCA
DHCR7	TCCACAGCCATGTGACCAATGC	CGAAGTGGTCATGGCAGATGTC
XBP1	CTGCCAGAGATCGAAAGAAGGC	CTCCTGGTTCTCAACTACAAGGC
CEBPB	AGAAGACCGTGGACAAGCACAG	CTCCAGGACCTTGTGCTGCGT
CHOP (DDIT3)	GGTATGAGGACCTGCAAGAGGT	CTTGTGACCTCTGCTGGTTCTG
Noxa (PMAIP1)	CTGGAAGTCGAGTGTGCTACTC	TGAAGGAGTCCCCTCATGCAAG
GRP78 (HSPA5)	CTGTCCAGGCTGGTGTGCTCT	CTTGGTAGGCACCACTGTGTTC

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
