# Peer review of "Activation of LXR Receptors and Inhibition of TRAP1 Causes Synthetic Lethality in Solid Tumors"

_cancers, 2019, doi:10.3390/cancers11060788_

Round 1

Reviewer 1 Report

In  line 107 cell death “induction” is missing

Figure 2D. the cell viability in U87MG control groups is not the same. Comparison of treatment groups should be done on relative values (control = 100%) for better understanding of the influence of LDL.

Figure 2C. fraction of cell viability > 100%? Please comment on the fact that viability is enhanced under combination therapy with high dose MEV

In vivo experiments: body weight curves and Kaplan Meier curves are mandatory for toxicity evaluation. Please add those. Student´s t test is not the appropriate statistical test for the comparison of multiple groups. Please elaborate why you used this or use another test suitable for multiple groups.

The data set would benefit tremendously from a quantitative analysis of the IHC data.

Can you elaborate on the role of cholesterol levels in the in vivo system? It would be beneficial to have some in vivo data indicating that a similar mechanism is induced in vivo as meticulously shown in vitro.

Taken together, it is a very interesting and well written paper. The authors emphasize more than one time how important the in vivo results are in this context. Thus, it is essential that they translate some of their mechanistic findings from in vitro to in vivo.

Author Response

Uploaded file.

Reviewer 2 Report

The authors' findings support the role of TRAP1 in the suppression of the cholesterol synthesis pathway.

This will fits  with literature data highlighting the role of lipid metabolism pathway in brain cancer (e.g. Abate M et al., Sci Rep. 2017 Oct 26;7(1):14123) and in its malignancy.

Concerning this aspect however I feel the manuscript must be reshaped in order to point the attention on GBM and not also in other cancers whose etiopathogenesis differ from GBM.

Throughout the manuscript indeed, the authors use different cell models of cancer pathology and experimental setting (see for example the different technical approaches, cell lines and time setting showed in figure 3). 

In view of main findings in Glioblastoma, the authors should focus their attention exclusively on GBM cell lines and GBM xenograft and move the supporting data on HCT116 in supplementary informations.

Moreover as the entire work concerns the TRAP1role,  it is not sufficient to declare that TRAP1 is increased in cancer  cells. The first figure should be the overexpression of TRAP1 on the primary and commercial cell lines used and if it is possible on GBM tissues from some patients' surgical resection. 

Finally, the effect on combined treatment on NHA astrocyte as normal counterpart should be verified.

Author Response

Uploaded file

Round 2

Reviewer 1 Report

The authors addressed thoroughly all remarks from the reviewers and the paper gained thereby additional value

Reviewer 2 Report

The manuscript has been nicely improved. I have no other comments and I should suggest its acceptance in the present form.